# Cerebral Diseases in Liver Transplant Recipients: Systematic Review of Clinical Evidence

**DOI:** 10.3390/jcm11040979

**Published:** 2022-02-13

**Authors:** Paula Dudek, Paweł Andruszkiewicz, Remigiusz Gelo, Rafael Badenes, Federico Bilotta

**Affiliations:** 12nd Department of Anesthesiology and Intensive Care, Medical University of Warsaw, 02-097 Warsaw, Poland; paula.dudek@wum.edu.pl (P.D.); paul.andruszkiewicz@gmail.com (P.A.); remekgelo@poczta.onet.pl (R.G.); 2Department of Anesthesiology and Surgical-Trauma Intensive Care, Hospital Clínic Universitari de Valencia, University of Valencia, 46010 Valencia, Spain; 3Department of Emergency Medicine, Anesthesiology and Intensive Care, Policlinico Umberto I, “Sapienza” University of Rome, 00161 Rome, Italy; bilotta@tiscali.it

**Keywords:** liver transplantation, early postoperative complications, long-term follow-up complications, postoperative neurological complications, intracranial hemorrhage, postoperative cerebral infection after liver transplant

## Abstract

This systematic review presents clinical evidence on early and long-term cerebral diseases in liver transplant recipients. The literature search led to the retrieval of 12 relevant studies. Early postoperative cerebral complications include intracranial hemorrhage associated with a coexisting coagulopathy, perioperative hypertension, and higher MELD scores and is more frequent in critically ill recipients; central pontine and extrapontine myelinolysis are associated with notable perioperative changes in the plasma Na+ concentration and massive transfusion. Long-term follow-up cerebral complications include focal brain lesions, cerebrovascular diseases, and posterior reversible encephalopathy; there is no proven relationship between the toxicity immunosuppressive drugs and cerebral complications. This SR confirms a very low incidence of opportunistic cerebral infections.

## 1. Introduction

Cerebral diseases—including cerebral hemorrhage, cerebrovascular, encephalopathies, etc.—after a liver transplant (LT) can complicate the early postoperative period, before hospital discharge, or occur can during long-term follow-up and are reported in 15–71% of patients [1,2,3,4,5]. Something of note in liver transplant patients is that neurological events are more frequent than after other types of solid-organ transplantation [6]. The etiology of this is multifactorial and is partially related to the fragile preoperative clinical condition of LT recipients (malnutrition, coagulopathy, multi-organ dysfunction, pre-LT encephalopathy, etc.) and to the complexity of the surgical procedure (major intraoperative hemorrhage with hemodynamic instability, major fluid shifts, etc.) and of the postoperative course [7,8]. Moreover, liver failure might lead to astrocyte metabolism disturbances, abnormalities in blood–brain barrier function, and the impairment of cerebral autoregulation due to cerebrovascular dilatation (during LT and after graft reperfusion) [9,10,11]. During long-term management, LT recipients are also exposed to the adverse effects of immunosuppression (diabetes, hypertension, hypercholesterolemia, impairment of renal function), which are related to an increased risk of cerebrovascular events in these patient populations [12]

Hence, patients who have undergone liver transplantation are at an increased risk for early and long-term cerebral complications; nevertheless, a systematic review (SR) that specifically addresses related clinical evidence is lacking.

The aim of this SR is to present clinical evidence on the early and long-term cerebral diseases present in LT recipients and to outline a dedicated work up for the perioperative management and long-term follow up of these patients.

## 2. Materials and Methods

A systematic literature search of PubMed and EMBASE was performed in accordance with the PRISMA (Preferred Reporting Items for Systematic Reviews and Meta-analyses) statement recommendations, and the study was registered in the Open Science Framework Registry (OSF registration DOI: 10.17605/OSF.IO/7CFYE) [13]. Clinical literature (prospective clinical trials, observational studies) was searched using the following search terms: “intracranial hemorrhage”, “cerebrovascular diseases”, “intracranial aneurysm”, “acute ischemic stroke”, “posterior reversible encephalopathy syndrome”, “central pontine and extrapontine myelinolysis”, “brain tumors”, “brain abscess” AND liver transplant. The group of selected key words was extended by screening the references of the included studies to find possible synonyms. The following filters were used: clinical studies; published between 2000 and October 2021; full-length articles (no abstracts); and published in English. After hand searching for articles and the revision of the full text, duplicates and irrelevant articles were eliminated. The details of the studies were recorded using a data-extraction form. Titles, abstracts, or both of studies retrieved using the search strategy and those from additional sources were screened independently, and the full text of potentially eligible studies was retrieved and assessed independently for eligibility. Disagreement over eligibility was resolved through open discussion. All of the trials were assessed to determine the risk of according to the Cochrane Collaboration’s criteria (http://handbook.cochrane.org accessed on 20 September 2020) Table 1.

## 3. Results

A total of 1715 articles were retrieved using the listed keywords. After screening for eligibility, 1703 articles were excluded, and 12 articles were selected (Figure 1) and were categorized into the two groups: early postoperative and long-term follow-up cerebral complications. Early postoperative cerebral complications included intracranial hemorrhage (ICH) and central pontine and extrapontine myelinolysis (CPM and EPM). Long-term follow-up cerebral complications included focal brain lesions, cerebrovascular diseases, and posterior reversible encephalopathy (PRES).

Evidence supported by a larger number of recruited patients will be displayed first.

### 3.1. Early Postoperative Cerebral Complications

This chapter will report on the clinical evidence reporting complications occurring during the early postoperative phase after LT (i.e., before postoperative discharge).

#### 3.1.1. Intracranial Hemorrhage

The literature search provided insight into the incidence and risk stratification of ICH in patients after LT. The clinical characteristics of the patients, possible risk factors, and the relationship between arterial blood pressure and postoperative ICH in LT recipients were evaluated in three retrospective cohort studies that enrolled a total of 2506 patients [14,15,16]. These studies report consistent evidence on the incidence of postoperative ICH during the observational period, which ranged between 30 days to 12 months, with figures ranging from 2% to 6.5%. Additionally, regarding the timing of ICH after LT, the three studies provide consistent indications, reporting that the highest risk takes place during the first two weeks after the procedure (Table 2). Two of the three studies reported low platelet counts (≤3 × 109/L) associate with an increased risk of ICH. Similarly, an increased intraoperative mean arterial pressure (MAP) ≥105 mmHg for 10 min or longer, a greater increase in pre- to posttransplant systolic blood pressure, a lower pretransplant serum fibrinogen level, and a higher pretransplant serum bilirubin level were associated with higher incidences of posttransplant ICH (Table 2) [14,15]. Patients with confirmed ICH had higher Model for End-Stage Liver Disease (MELD) scores and were more likely to have preoperative encephalopathy, be on vasopressors, be respirator-dependent, and have hemodialysis [14,15].

#### 3.1.2. Central Pontine and Extrapontine Myelinolysis

The prevalence and possible risk factors of CPM and EPM occurring after LT were evaluated in four retrospective studies that enrolled a total of 3622 patients [17,18,19,20]. In these studies, patients who developed CPM/EPM were compared to the transplanted controls without myelinolysis. The incidence of CPM/EPM was 0.88–3.5%. The clinical manifestation of CPM/EPM in most cases occurred within 2 weeks after LT (ranging between the third postoperative day and 2 months after LT). Remarkable changes in intra- and postoperative plasma Na+ concentration and large blood derivatives (RBC, FFP, etc.) that need to be transfused are predictors of CPM/EPM after LT (Table 2). In three of the four studies, preoperative hyponatremia was associated with CPM/EPM after LT [17,18,20]. Of the three studies that reported data on MELD or MELD-Na+ scores, an association with an increased risk of CPM/EPM was proven in two [17,18]. Despite previous evidence suggesting a possible relationship between the plasma concentration of immunosuppressive drugs above the therapeutic window and CPM/EMP, the three trials included in the present SR that recorded this variable do not confirm this association [17,19,20]. Hypercholesterolemia has also been suggested as a possible risk factor for CPM/EPM in a single study [18].

### 3.2. Long-Term Follow-Up Cerebral Complications

This paper will describe studies that describe complications shown to occur after LT in the late postoperative phase.

#### 3.2.1. Focal Brain Lesions

Focal brain lesions (cerebral abscess, brain lymphoma, etc.) have been reported to complicate the long-term follow-up of LT and have a higher incidence than they do in matched controls [21]. A large retrospective study included in this SR that presented data from 62,405 LT recipients confirms that primary central nervous system lymphoma (PCNSL) is more frequent among LT recipients than in a matched control population who did not receive LT but that it is less frequent than it is among kidney transplant recipients [21]. Of note, among LT recipients, PCNSL is associated with increased rates of graft failure/re-transplantation and higher mortality (Table 2). According to the same study, LT recipients who were seronegative for Epstein–Barr virus before surgery had a higher PCNSL incidence than seropositive recipients did.

#### 3.2.2. Cerebrovascular Diseases 

Several aspects of cerebrovascular diseases in LT recipients have been investigated, including prevalence and related risk factors [22,23,24].

In a retrospective cohort study that presented data from 1920 LT recipients and compared it to data from 24681 healthy adults, it was reported that the prevalence of cerebral aneurysms is similar in the two studied groups (3.1% vs. 3.8%), but the localization differed: the anterior communicating artery and superior cerebellar artery aneurysms were more frequent in LT recipients [22]. The authors highlight that aneurysms with this localization have a higher risk of rupture. Cerebral arteriovenous malformations were more frequent in LT recipients (0.26% vs. 0.06%), but no differences were observed in terms of cavernous malformation (Table 2).

The rupture risk of diagnosed an intracranial aneurysm and, connected with this, the incidence of a subarachnoid hemorrhage (SAH) and hemorrhagic stroke after LT was investigated in a retrospective cohort study based on 3544 LT recipients [23]. The prevalence of unruptured cerebral aneurysms in the LT recipients was 4.63%, and was more predominant in women with a history of hypertension and did not differ across the etiologies of cirrhosis and its severity or a history of diabetes, dyslipidemia, and smoking. The incidence of SAH in patients presenting with an unruptured intracranial aneurysm at the time of LT was 0.68% at the one-year follow-up. The presence of an unruptured intracranial aneurysm was not a risk factor for SAH, hemorrhagic stroke, or mortality after LT, especially when considering that during the median follow-up period of 4.5 years, only one of the 147 patients with an unruptured intracranial aneurysm developed symptomatic SAH. Cirrhosis severity/MELD score, thrombocytopenia (≤50.000 dL^−1^), inflammation defined as a C-reactive protein concentration > 1.8 mg/dL, and history of SAH were identified as risk factors for one-year hemorrhagic stroke after LT.

To test whether LT might have an effect on the incidence of cerebrovascular events, i.e., stroke or transient ischemic attack (TIA), a retrospective cohort of 313 LT recipients were studied according to their PROCAM Stroke scores and their 20-year follow-up [24]. In this study population, the risk of cerebrovascular events in LT recipients was 3.5-fold higher that it was during the first 10 years and 2-fold higher in the second decade after LT than it was in “the standard population” (Table 2).

#### 3.2.3. Posterior Reversible Encephalopathy Syndrome

Clinical features and potential risk factors for PRES were analyzed in a retrospective cohort study that reported data from 1923 LT recipients [25]. Out of these patients, 19 (1%) were diagnosed radiologically with PRES. In 16 (84%) of these cases, PRES occurred within 3 months after LT, with the mean time between LT and PRES diagnosis of 31 days. The most common clinical manifestation was seizures, and six (31%) patients with PRES also had ICH, which was associated with coagulopathy (INR >2 and/or PLT count < 100 × 103/L) in all cases. Among patients presenting with PRES, a large proportion received a LT due to alcoholic liver disease and frequently presented concomitant infections/sepsis. However, no differences were observed in terms of acute cellular rejection before PRES, and it was more frequent in patients who developed PRES (56% vs. 19%), but no differences in symptoms were observed before PRES started. No differences in operative-related data, tacrolimus levels, and electrolyte concentration between the groups were noticed (Table 2). Although 31% of the patients with PRES had residual lesions during MR imaging, none of the survivors showed any residual neurologic deficits.

## 4. Discussion

This SR reports clinical evidence on cerebral diseases in LT recipients. These patients can develop cerebral complications in the early postoperative phase after LT because of the underlying associated disturbances (i.e., coagulation abnormalities, endothelial frailty, chronic encephalopathy, etc.) or in the long-term follow-up, most of which are the consequence of the long-standing use of immunosuppressive therapy. In the early postoperative phase, the risk of ICH increases, as does the coexistence of coagulopathy with perioperative hypertension and higher MELD scores and is more likely to occur in critically ill recipients (preoperatively on vasopressors, who are ventilator-dependent, and who are on hemodialysis). The incidence of CPM/EPM is associated with notable perioperative changes in the plasma Na^+^ concentration and the massive transfusion of blood products. During the long-term follow-up after LT, there was not an excess of intercranial aneurysms; however, cerebrovascular events (i.e., stroke, TIA) were more frequent. From the pretransplant variables, the total plasma bilirubin level and low-plasma total cholesterol level were associated with increased incidences of ICH and CPM/EPM, respectively. There is no proven relationship between immunosuppressive drug toxicity and cerebral complications that might be associated with a high tacrolimus plasma level.

This SR was conducted in accordance with the PRISMA guidelines. Given the fact that the immunosuppression regimen that is currently prescribed after LT has changed (switching from triple-drug to single-drug immunosuppression, minimizing the dosages of calcineurin inhibitor drugs, new classes of immunosuppressants), the literature search was limited to the year 2000 and onwards in order to present more recent data on the actual picture of neurological complications in LT recipients. These SR confirm a very low incidence of opportunistic cerebral infections (limited to case reports) and a high prevalence of cerebrovascular complications after LT as reported by Vizzini and colleagues [4].

Given the severity and the decreased graft and patient survival as well as the cost of the prolonged hospital stay arising from cerebral complications after LT, effort should be made to detect and prevent them as soon as possible. Based on the published data included in this SR, the following are recommended for post-LT management: (a)There should be prompt correction of existing coagulopathy to minimize the risk of ICH, especially in patients with diagnosed PRES;(b)Arterial blood pressure should be closely monitored and hypertension should be treated in the perioperative period, as hypertension has been proven to be associated with ICH;(c)As the most ICH occur within first two weeks, patients who have undergone LT should receive acerebral CT scan immediately should new neurological deficits occur;(d)Preoperative natremia should be carefully monitored, and diuretics should be used cautiously;(e)Perioperative hyponatremia should be treated slowly, and the correction rate should not exceed 15 mmol/L/24 h or 18 mmol/L/48 h;(f)Surgical techniques to minizime intraoperative blood loss should be used to reduce the risk of CPM/EPM;(g)of the plasma total cholesterol level should be assessed to identify LT recipients who are at an increased risk of CMP/EPM;(h)The plasma level of immunosuppressants should be tightly controlled to avoid neurotoxicity;(i)As the risk of cerebrovascular events is notably higher after LT than expected, long-term surveillance and active screening should be implemented;(j)Because of intracranial aneurysms locations that are at higher risk of rupture, a close observation and aggressive management strategy are suggested [26];(k)LT recipients who are seronegative for Epstein–Barr virus before surgery are at risk of PCNSL and should be monitored for this complication.

This SR has several limitations, including the methodology, which relied on a literature search that was limited to two databases (PubMed and EMBASE); however, it is appropriate to note here that as the most comprehensive databases, the risk of omitting major information is limited. Among the possible limitations, it should be mentioned that a limited selection of key words was selected for the literature search. This might have prevented studies that are potentially related to the topic from being reached. However, even though we acknowledge this possible risk it is unlikely that relevant studies were missed because we exercised clinical competence through searching the references of the selected papers. Another possible limitation refers to the exclusion of studies in the form of case reports, which might have prevented us from reporting rare complications, i.e., opportunistic brain abscess in this SR.

## 5. Conclusions

In conclusion, LT recipients show a specific path of early and long-term follow-up cerebral complications. In these patients, it is necessary to design a dedicated diagnostic work-up right in the immediate postoperative phase and to complete a tailored and appropriate follow-up. Several risk factors identify an increased risk for postoperative cerebral complications, and these should be ruled out. Future studies should address purposeful therapeutic strategies to prevent these complications.

## Figures and Tables

**Figure 1 jcm-11-00979-f001:**
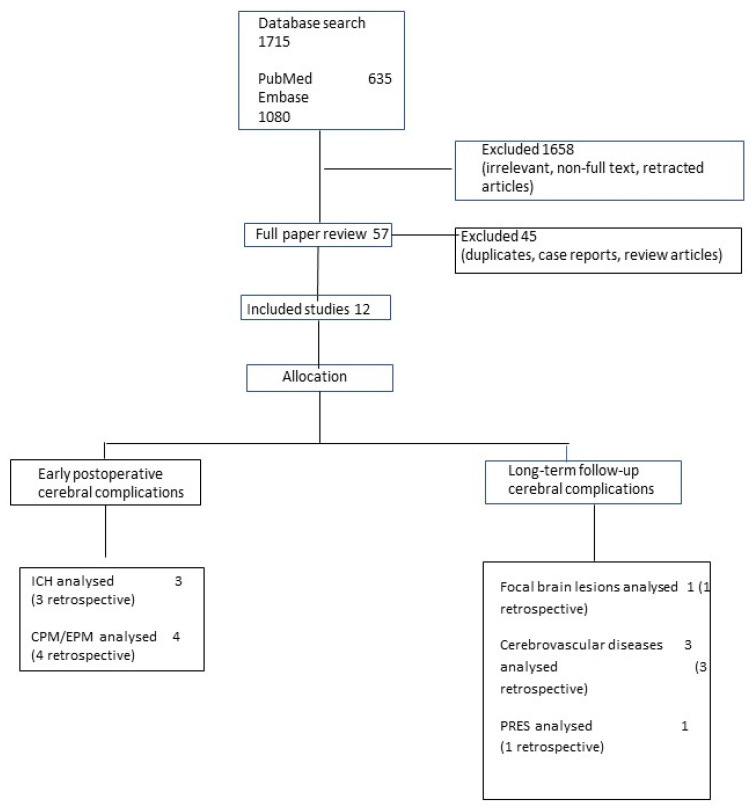
ICH, intracranial hemorrhage; CPM/EPM, central pontine and extrapontine myelinolysis; PRES, posterior reversible encephalopathy.

**Table 1 jcm-11-00979-t001:** Risk of bias of the selected studies.

I.D	Bias Due to Confounding	Bias in Selection of Participants into the Study	Bias in Measurement of Interventions	Bias Due to Departures from Intended Interventions	Bias Due to Missing Data	Bias in Measurement of Outcomes	Bias in Selection of the Reported Result
Gao et al. [14]	U	M	M	M	L	M	L
Gallagher et al. [15]	U	M	M	M	L	M	L
Wang et al. [16]	U	M	M	M	L	M	M
Morard et al. [17]	U	M	M	H	M	M	L
Lee et al. [18]	U						
Crivellin et al. [19]	U	H	M	H	M	H	M
Yu et al. [20]	U	H	M	H	M	H	M
Mahale et al. [21]	U	M	L	L	M	L	L
Chung et al. [22]	U	M	M	L	L	L	L
Kwon et al. [23]	H	H	H	M	M	L	L
Schoening et al. [24]	U	H	L	M	H	L	L
Cruz et al. [25]	U	H	M	L	L	M	L

L, low risk of bias; M, moderate risk of bias; S, serious risk of bias; C, critical risk of bias; U, unclear risk of bias.

**Table 2 jcm-11-00979-t002:** Characteristics of studies included in this SR.

Study	Study Design *N*	Primary End Piont	Mortality	Key Points
Gao et al. [14]	Retrospective *N* = 1836	ICH	For 30 days: 48.3%	Identified risk factors for posttransplant ICH: - Intraoperative MAP ≥ 105 mmHg for ≥10 min; - Intraoperative PLT counts ≤ 30 × 10^9^/L; - Preoperative total bilirubin level ≥ 7 mg/dL.
Gallagher et al. [15]	Retrospective *N* = 595	ICH: Intraparenchymal hemorrhage (IPH) and extra-axial hemorrhage (EAH)	For 30 days: 33.3% for IPH 12.% for EAH	Identified risk factors for posttransplant ICH: - Female sex; - Greater increase in pre- to posttransplant SBP; - Lower pretransplant serum fibrinogen level; - Higher pretransplant total bilirubin level; - Higher MELD scores.
Wang et al. [16]	Retrospective *N* = 75	ICH	80%	Identified risk factors for posttransplant ICH: - Greater intraoperative blood transfusion volume; - Intraoperative hypotension.
Morard et al. [17]	Retrospective *N* = 1378	Possible risk factors for CPM and EPM	For 1-year 63%	Possible causes of CPM/EPM: - Low (<130 mmol/L) and very low (<125 mmol/L) plasma Na^+^ concentration before LT; - Increasing of Na^+^ ≥12 mmol/L in the postoperative period; - Transfusion of ≥4 platelet units, of ≥12 FFP; - Hemorrhagic surgical complications. The association of ≥3 of these risk factors was strongly associated with CPM/EPM occurrence.
Lee et al. [18]	Retrospective *N* = 1247	Possible risk factors for CPM and EPM	N/A	Possible causes of CPM/EPM: - Higher MELD-Na^+^score; - Preoperative hyponatremia and hypocholesterolemia; - Greater changes in plasma Na+ and K^+^ concentration during LT; - Greater volume of transfused blood components and crystalloids during LT. No differences in the duration of LT or underlying liver disease type of immunosuppressant between the groups.
Crivellin et al. [19]	Retrospective *N* = 997	Prevalence and possible risk factors for CPM and EPM	For 1 year 9.1% For 5 years: 18.8%	Possible causes of CPM/EPM - Higher variations in intra- and perioperative plasma Na^+^ concentration within 24 h post-LT No difference in the preoperative plasma Na^+^, K^+^ concentration, MELD-Na^+^, and underlying liver disease between the groups.
Yu et al. [20]	Retrospective *N* = 142	Possible risk factors for CPM	Overall: 100%	Possible causes of CPM: - Preoperative hyponatremia; - Greater changes in plasma Na^+^ concentration during 48 h post-LT; - Plasma osmolality post-LT; - Duration of LT.
Mahale et al. [21]	Retrospective *N* = 288,029 solid organ recipients with 62,405 (21.7%) *LT*	The incidence of PCNSL and systemic NHL in transplant recipients.	N/A	Higher incidence of PCNSL after LT than in non-transplanted population. PCNSL incidence after LT is lower than after other organ transplantations. LT recipients seronegative for Epstein–Barr virus before LT had higher PCNSL incidence than seropositive recipients. PCNSL increased mortality, the incidence of graft failure, and retransplantation rates.
Chung et al. [22]	Retrospective *N* = 1920	The prevalence of cerebral aneurysms, cerebral arteriovenous malformation, and cavernous malformation	N/A	No differences in overall incidence of cerebral aneurysms between LT recipients and control groups. Different distributions of cerebral aneurysms according to location in the groups. Higher incidence of cerebral arteriovenous malformation in LT recipients than in the control group. No difference in the occurrence of cavernous malformation between the groups.
Kwon et al. [23]	Retrospective *N* = 3527	1 year symptomatic SAH	For 1 year: no differences between patients with and without asymptomatic unruptured intracranial aneurysm.	1 year SAH incidence in patients with unruptured intracranial aneurysm after LT was 0.68%. Unruptured intracranial aneurysm is not a risk factor for SAH, hemorrhagic stroke, or mortality after LT. Identified risk factors for posttransplant 1 year hemorrhagic stroke: - Higher MELD scores; - PLT ≤ 50 × 10^9^/L; - CRP ≥ 1.8 mg/dL; - History of SAH.
Schoening et al. [24]	Retrospective *N* = 313	The incidence of cerebrovascular events (TIA or stroke) 6 months and 10 and 20 years after LT using PROCAM Stroke score.	N/A	Higher cerebrovascular risk in LT recipients than expected based on PROCAM Stroke score compared to the standard population.
Cruz et al. [25]	Retrospective *N* = 1923	Risk factors of PRES	0%	- Seizure is the most common clinical manifestation; - 31% PRES cases associated with ICH; - Risk factors; - coagulopathy, ALD, infection.

ICH, intracranial hemorrhage; MAP, mean arterial pressure; PLT, platelets; SBP, systolic blood pressure; MELD, model for end-stage liver disease; SAH, subarachnoid hemorrhage; N/A, not applicable; LT, liver transplant; CPM, central pontine myelinolysis; ALD, alcoholic liver disease; EPM, extrapontine myelinolysis; TIA, transient ischemic attack; PCNSL, primary central nervous system lymphoma; NHL, non-Hodgkin lymphoma.

## Data Availability

Not applicable.

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
