# Peer review of "Cerebral Diseases in Liver Transplant Recipients: Systematic Review of Clinical Evidence"

_jcm, 2022, doi:10.3390/jcm11040979_

Round 1

Reviewer 1 Report

Interesting review paper. This paper is mostly well organaized and well written. 

I have only one question to ask on serch terms. The authors used the term: "intracranial hemorrhage", " cerebrovascular disease", "intracranial aneurysm", "acure ischamic stroke", and so on (line 54-57).

Why were these terms selected for this systematic review? Wasn't there a risk to miss some important studies which utilized the term such as "cerebral infarction" or "cerebral aneurysm"? In other words, did these terms cover all synomym?

Reviewer 2 Report

Although this article is described as a systematic review it is only a literature review of postoperative neurologic complications after liver transplantation and hence, it does not bring anything new to the literature. The format of the review is more appropriate as a textbook chapter. I recommend the authors to focus on one of the key points of neurologic postoperative complications and perform a systematic review on that matter - eg. Central pontine demyelination. 

Round 2

Reviewer 2 Report

The few changes you have made did not improved the overall quality of the manuscript and in my opinion it suffers from the same weaknesses already highlighted in my previous report.

I appreciate your work, however, I believe the changes made are insufficient and far more minor than the weaknesses already presented.

Since there are minor significant improvements in this sense, I can only confirm, albeit with regret, my judgment. I hope you take the revision more in-depth to enhance the quality of your manuscript. You need to do major revision.

I wish you all good work.